

# Multi-label classification for image tamper detection based on Swin-T segmentation network in the spatial domain

Li Li[1], Kejia Zhang[1], Jianfeng Lu[1,2] and Shanqing Zhang[1]

[1] School of Computer Science and Technology, Hangzhou Dianzi University, Hangzhou, Zhejiang, China
[2] Shangyu Institute of Science and Engineering, Hangzhou Dianzi University, Shaoxing, Zhejiang, China

## ABSTRACT

The majority of deep learning methods for detecting image forgery fail to accurately detect and localize the tampering operations. Furthermore, they only support a single image tampering type. Our method introduces three key innovations: (1) A spatial perception module that combines the spatial rich model (SRM) with constrained convolution, enabling focused detection of tampering traces while suppressing interference from image content; (2) A hierarchical feature learning architecture that integrates Swin Transformer with UperNet for effective multi-scale tampering pattern recognition; and (3) A comprehensive optimization strategy including auxiliary supervision, self-supervised learning, and hard example mining, which significantly improves model convergence and detection accuracy. Comprehensive experiments are performed on two established datasets; namely MixTamper and DocTamper with 19,600 and 170,000 images, respectively. The experimental findings demonstrate that the proposed model enhances the IoU index by 13% compared to the leading algorithms. Additionally, it can accurately detect multiple tampering types from a single image.

# INTRODUCTION

As computer vision rapidly advances, numerous image tampering techniques are continuously emerging. Image forgery entails intentionally altering pixels in digital images to mislead perceptions of authenticity, often through techniques like copy-move forgery, where a part of the image is replicated within itself; splicing forgery, which inserts content from another image; smearing forgery, which distorts or blurs pixels to obscure details; erasing forgery, by removing or covering content to alter the image's appearance; text-generating forgery, through artificially creating text to modify a document's content; and hybrid forgery, where multiple forgery types are combined to increase complexity and detection difficulty, as shown in Fig. 1, with tampered areas marked by red boxes. Detecting these forgeries relies on analyzing inconsistencies in texture, lighting, edges, and other image features to reveal tampering traces.

Corresponding author
Shanqing Zhang,
sqzhang@hdu.edu.cn

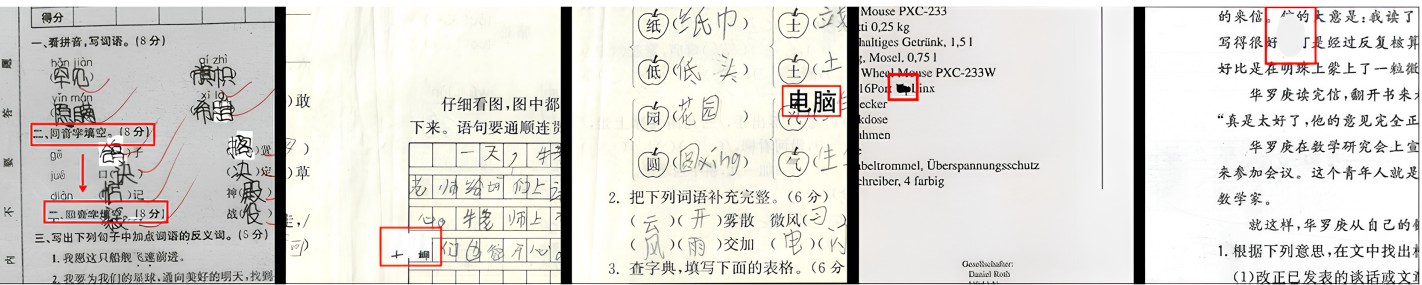

**Figure 1 Examples of forgery types with tampered areas highlighted in red.**

In order to detect the specific ways of image tampering, researchers have proposed image tampering detection algorithms. Most of the existing image tampering detection methods typically rely on various convolutional neural network (CNN)-based feature extractors to detect the unique inconsistent traces resulting from tampering (*Liu et al., 2023*; *Li et al., 2023*; *Guillaro et al., 2023*; *Wang et al., 2022*). CNNs are generally better suited for tasks that involve semantic understanding, such as object detection, as they excel at identifying meaningful features within objects. However, tampering traces often appear around the periphery of objects and are non-semantic in nature. To effectively identify the inconsistencies between traces and original regions, it is crucial to compare relationships across different areas. Given the need to capture feature differences between manipulated and authentic regions, we argue that self-attention mechanisms offer a more effective solution. Self-attention mechanisms can explicitly model relationships between any areas, regardless of their visual semantic relevance, making them particularly effective for capturing dependencies between non-adjacent regions. Additionally, the sliding window mechanism significantly reduces the computational overhead caused by processing the entire image as input. Therefore, Swin Transformer (Swin-T) (*Liu et al., 2021b*) emerges as an ideal choice for this task.

Spatial domain information in images is diverse and rich, encompassing content details, noise, traces, texture features, and artifacts resulting from various attacks, such as compression and geometric distortions. As a result, the task of detecting image tampering must direct the model to concentrate on the boundaries of altered areas and understand their distribution to improve detection performance. We propose a spatial domain sensing module that directs the model's focus towards the periphery of the tampered regions, further distinguishing it from traditional object detection tasks.

In datasets, the proportion of tampered areas compared to the entire image area varies greatly among different types of tampering. For example, copy-move forgeries often occupy a large portion of the image, whereas smearing or erasing forgeries cover only a small number of pixels. This discrepancy poses a challenge for the design of our multi-label classification network. Swin-T (*Liu et al., 2021b*) adopts a hierarchical structure where the resolution of feature maps progressively decreases across layers, while the semantic

information of the features is gradually enhanced. This multi-scale representation is highly effective for capturing details of varying sizes. To fully leverage these features, we propose an improved feature pyramid structure.

Existing studies have achieved an excellent accuracy for image tampering detection and have shown good performance for tampering classification tasks. However, the accuracy of the algorithms with respect to tamper localization still needs some improvements. Many image tampering detection algorithms can effectively detect the approximate region of forgery, but it fails to determine exactly the boundary of the tampering region. In particular, the sensitivity of the existing algorithms appears to be insufficient when confronted with subtle traces of tampering. In addition, due to the design principle of the multi-classification module, image forgery detection algorithms can only detect a single type of tampering from the image based on the prediction results. This means that when multiple tampering types are presented in a single image, the classifier tends to select the category with more tampering for recognition. Therefore, the existing image forgery detection algorithms are limited to deal with the complex situations of multiple tampering types.

The key innovations presented in this article can be summarized as follows:

(1) We improve UperNet (*Xiao et al., 2018*) and design a Multi-resolution Fusion UperNet network (MRF-UperNet) for segmentation based on Swin-T. The network is then used for multi-label classification tampering detection, which utilizes the unique self-attention mechanism and multi-resolution fusion structure in the network to achieve more accurate detection of document image tampering details.

(2) The spatial domain sensing module is innovatively designed in the first layer of the network. This module incorporates constrained convolution, Spatial Rich Model (SRM) convolution and ordinary convolution. This approach enhances the network's sensitivity to tampering features while minimizing the impact of irrelevant features, thereby improving detection accuracy.

(3) The introduction of auxiliary detection heads for co-training helps the network to better cope with more complex patterns of document tampering.

(4) To address the problems of sample imbalance and differences in training difficulty, we introduce a mixed loss function and enhance the proposed model using hard sample mining and self-supervised dataset augmentation to boost tampering detection performance.

## RELATED WORK

### CNN and variants

CNNs are among the most iconic neural network types, extensively utilized in various image processing tasks. They generally feature a hierarchical architecture comprising convolutional layers, batch normalization layers, fully connected layers, and additional layers. In the literature, there are many advanced and efficient neural network structures that have been introduced, *e.g.*, AlexNet (*Krizhevsky, Sutskever & Hinton, 2012*), fully

connected network (FCN) (*Long, Shelhamer & Darrell, 2015*), ResNet (*He et al., 2015*), HR-Net (*Sun et al., 2019*), and UperNet (*Xiao et al., 2018*). *Bondi et al. (2017)* trained a CNN to extract features specific to the device that captured the image. The algorithm determines that an image has been tampered if an image includes two or more features of the capturing device. *Qian et al. (2021)* introduced a CNN model for image tampering detection that leverages an image preprocessing algorithm. They combined the traditional support vector machine and machine learning methods to extract the features from qualification certificates for image tampering detection. To further enhance the detection accuracy of the convolutional network, the researchers adjusted the network structure by deepening the layers of the convolutional network, based on residuals, and further adding attention mechanisms. In this article, we utilize FCN (*Long, Shelhamer & Darrell, 2015*) to process input images of any size and eventually restore them to their original dimensions through upsampling, to further assist the network in achieving pixel-level tampering classification training. Furthermore, we improve pyramid structure of the UperNet (*Xiao et al., 2018*) network to fully integrate image features at different resolutions (hereafter referred to as MRF-UperNet).

## Image manipulation detection/localization

Initial studies mainly focused on detecting particular types of manipulations, such as splicing (*Cozzolino, Poggi & Verdoliva, 2015b*; *Huh et al., 2018*; *Kniaz, Knyaz & Remondino, 2019*), copy-move (*Cozzolino, Poggi & Verdoliva, 2015a*; *Rao & Ni, 2016*), and removal (*Zhu et al., 2018*). Nevertheless, the precise nature of the manipulation is frequently uncertain in the practical situations. This uncertainty has driven a growing body of research focused on developing methods for general manipulation detection. *Alshanbari (2021)* employed Discrete Wavelet Transform (DWT) decomposition for medical images. *González Fernández et al. (2018)* initially calculated the likelihood of each pixel being interpolated and then applied the Discrete Cosine Transform (DCT) on small blocks of the probability map to identify tampered regions within the image. PSCCNet (*Liu et al., 2021a*) extracts hierarchical features *via* a top-down approach and assesses whether an input image has been manipulated using a bottom-up approach. RRU-Net (*Bi et al., 2019*) integrates residual blocks within both the encoder and decoder to address the vanishing gradient issue in deep networks and employs recurrent units in the skip connections. MVSS-Net (*Dong et al., 2021*) uses multi-perspective feature learning to collect information from various perspectives and applies multi-scale supervision to enhance detection accuracy by combining features across multiple resolutions. *Zhou et al. (2023)* employ a contrastive learning approach to tackle the issue of data insufficiency in image tampering detection. *Ma et al. (2023)* enhanced the method at the patch level instead of the pixel or image level, utilizing Vision Transformer for benchmarking and proposing several evaluation metrics. *Qu et al. (2024)* classified tampered images into two categories based on foreground and background, facilitating the automatic and precise pixel-level labeling of numerous manually forged images obtained from the web. In this article, we employ a multi-class network structure for tampering detection, achieving precise localization by capturing spatial information.

## Transformer-based backbones

The Transformer architecture was first introduced in 2017. Its ability to effectively capture long-range feature dependencies quickly made it a dominant machine learning framework for natural language processing tasks (*Vaswani et al., 2017*). However, its application in the field of image processing is limited due to its huge computational complexity and inability to handle image information with high resolution. To decrease computational demands and fully leverage the benefits of the Transformer, some researchers innovatively introduced the Vision Transformer (ViT) (*Dosovitskiy et al., 2020*). ViT and its follow-ups (*Touvron et al., 2021*; *Yuan et al., 2021*; *Chu et al., 2021*; *Han et al., 2021*; *Wang et al., 2021*) adapted the Transformer architecture for computer vision, where its attention mechanism effectively captured long-range dependencies among different elements in a sequence, enhancing sequence processing.

In the context of image tampering detection, several recent Transformer-based approaches have shown promising results but face significant limitations. TransForensics (*Hao et al., 2021*) combines self-attention encoders with dense correction modules, achieving rich point interactions and multi-scale correction. However, it suffers from resolution constraints and high memory usage, limiting its practical application. ObjectFormer (*Wang et al., 2022*) introduces object-level attention and RGB-frequency fusion for global semantic understanding, but its limited frequency learning capability and sensitivity to compression artifacts restrict its performance in real-world scenarios. IML-ViT (*Ma et al., 2023*) employs a pure Vision Transformer backbone with edge supervision, offering high-resolution processing capabilities but at the cost of excessive computational requirements.

After comprehensive comparison of these approaches, we identified Swin Transformer (Swin-T) (*Liu et al., 2021b*) as the most promising foundation for our work. As a subsequent innovation following ViT, Swin-T introduces a sliding window mechanism that effectively balances global and local information processing while maintaining computational efficiency. Unlike TransForensics's memory constraints, ObjectFormer's limited frequency learning, and IML-ViT's computational overhead, Swin-T provides an optimal balance of feature extraction capability and practical efficiency. Building upon this foundation, we enhance Swin-T with our spatial domain sensing module and multi-resolution fusion structure, specifically designed for tampering detection while preserving its computational advantages. In this article, we employ a segmentation backbone network built on Swin-T.

## METHODS

In this study, we introduce a multi-classification tampering detection algorithm utilizing a spatial domain Swin-T segmentation network. The core structure of the proposed model is depicted in Fig. 2. The algorithm is composed of four modules: the spatial domain sensing module, the Swin-T backbone network module, the MRF-UperNet decoding network module, and the auxiliary detection head FCN module. The process is as follows: initially, the image is fed into the spatial domain sensing module for preprocessing, and the resulting output is processed through the Swin-T module, which performs feature

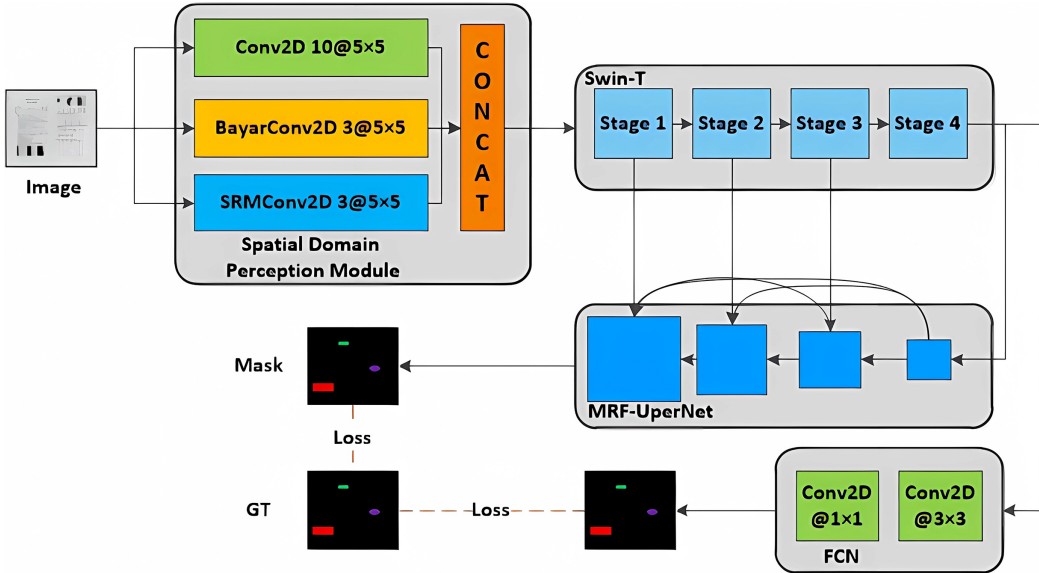

**Figure 2 Overall structure of the proposed tamper detection model.**

extraction across four stages. Subsequently, the features extracted are sent to the MRF-UperNet decoder network for multi-scale fusion. Ultimately, this yields the multi-classification tampering mask of the input image. Additionally, to expedite network convergence, we incorporate an auxiliary detection head FCN module, which receives output from the Swin-T and computes the joint loss function as an additional training branch.

Meanwhile, we implement a series of measures to optimize the network during its training process. Specifically, a self-supervised augmentation measure is employed to expand the document tampering dataset by generating samples possessing with more complex tampering types. Additionally, a hard sample mining strategy is employed to increase the sensitivity of the model to subtle tampering traces. Finally, a joint loss function is innovatively proposed to fully consider both the classification of different tampering types and the disparity between positive and negative instances. Consequently, the proposed model effectively overcomes the limited recognition capabilities of current state-of-the-art methods when dealing with complex tampering operations.

## Network design

### Spatial domain sensing module

The spatial image information is rich and diverse, not only contains content information, but also may be mixed with noise, edge details and potential tampering information. For those subtle tampering of document images that are difficult to recognize with the naked eye. Accurate detection based solely on the image's visual information is challenging, necessitating the detection and analysis of tampering information concealed within the image. To allow the network to grasp more spatial information, the spatial domain sensing module has been designed. This module applies three different convolutional operations:

SRMConv2D (*Zhou et al., 2018*), BayarConv2D (*Bayar & Stamm, 2018*) and classical Conv2D layers to the spatial information, and combine the extracted features by a simple concatenation operation.

SRMConv2D uses three SRM filters, whose weights are shown in Fig. 3. We froze the weights so that local noise features are extracted more efficiently by utilizing the difference between a pixel's actual value and its estimated value, obtained through interpolation from adjacent pixel values, thus detecting tampering artifacts that are difficult to detect in the RGB channels.

BayarConv2D achieves effective perception of image edge information and detection of tampering traces by introducing specific constraints in the convolution operation. These constraints are shown in Eq. (1), where $w_k$ denotes the *kth* convolution filter and the center value of the convolution filter is denoted by $(0,0)$. The constraints dictate that the center value of the convolutional filter must be $-1$, while the sum of the weights at all non-center positions must equal one. This design allows the convolutional layer to build up information with residuals to better capture edge details and potential tampering traces in the image.

$$\begin{cases} w_k(0,0) = -1, \\ \sum_{m,n\neq0} w_k(m,n) = 1, \end{cases} \tag{1}$$

### Backbone network Swin-T

The backbone network is employed to extract high-level features indicative of tampering. This subsection refers to different models in selecting the backbone network, and comparatively analyzes the efficacey of different backbone networks such as ResNet (*He et al., 2015*), U-Net (*Ronneberger, Fischer & Brox, 2015*), HRNet (*Sun et al., 2019*) and Swin-T (*Liu et al., 2021b*), and finally selects Swin-T based on the experimental effects and training speed. Then, we introduce the relevant principle of Swin-T network and its construction process.

#### Principle

Swin-T (*Liu et al., 2021b*) is an innovative visual Transformer, functioning as a convolutional neural network built upon the Transformer architecture. Compared to the traditional Transformer, Swin-T aims to reduce the computational burden on high-resolution image processing. It employs a staged image processing strategy, where a large-size input image is decomposed into multiple small-size image blocks, each of which is called a patch. Inside the patch, Swin-T performs self-attentive computation and establishes patch-to-patch dependencies with sliding window operations. Therefore, it significantly reduces the computational complexity.

Swin-T innovatively proposes two main modules:

(1) Module for changing sequence length

Swin-T can modify the module structure, increase the number of channels, and create a hierarchical design while reducing computational load through three operations: the patch partition layer, linear embedding layer, and patch merging layer. The patch partition layer

$$\frac{1}{4}\begin{bmatrix} 0 & 0 & 0 & 0 & 0 \\ 0 & -1 & 2 & -1 & 0 \\ 0 & 2 & -4 & 2 & 0 \\ 0 & -1 & 2 & -1 & 0 \\ 0 & 0 & 0 & 0 & 0 \end{bmatrix} \quad \frac{1}{12}\begin{bmatrix} -1 & 2 & -2 & 2 & -1 \\ 2 & -6 & 8 & -6 & 2 \\ -2 & 8 & -12 & 8 & -2 \\ 2 & -6 & 8 & -6 & 2 \\ -1 & 2 & -2 & 2 & -1 \end{bmatrix} \quad \frac{1}{2}\begin{bmatrix} 0 & 0 & 0 & 0 & 0 \\ 0 & 0 & 0 & 0 & 0 \\ 0 & 1 & -2 & 1 & 0 \\ 0 & 0 & 0 & 0 & 0 \\ 0 & 0 & 0 & 0 & 0 \end{bmatrix}$$

**Figure 3  Three SRM filter cores.**

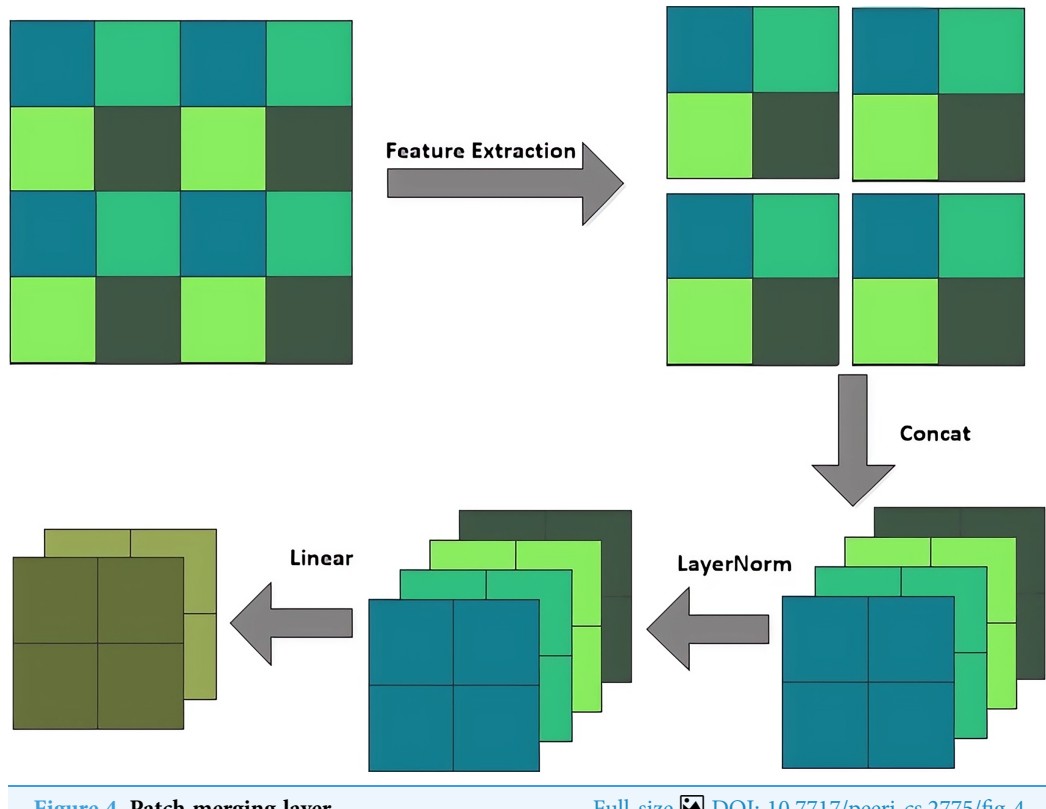

**Figure 4  Patch merging layer.**

segments and stacks the channel modules, the linear embedding layer alters the feature dimensions, and the patch merging layer functions similarly to a pooling layer. The patch merging layer executes a down-sampling operation prior to each processing step, by selecting elements in the direction of rows and columns through intervals and then recombining them into new patches, and hence feature resolution reduction can be achieved. Its structure is shown in Fig. 4. Unlike the traditional pooling operation, the Patch merging layer focuses more on feature selection and reorganization. So, it can extract and deliver key information more effectively.

(2) Multi-head attention module based on windows

It primarily comprises two types of modules: Window Multi-Head Self-Attention (W-MSA) and Shifted Window Multi-Head Self-Attention (SW-MSA). W-MSA begins by partitioning the feature map into non-overlapping windows and independently executing self-attention computations within each window, thereby significantly reducing

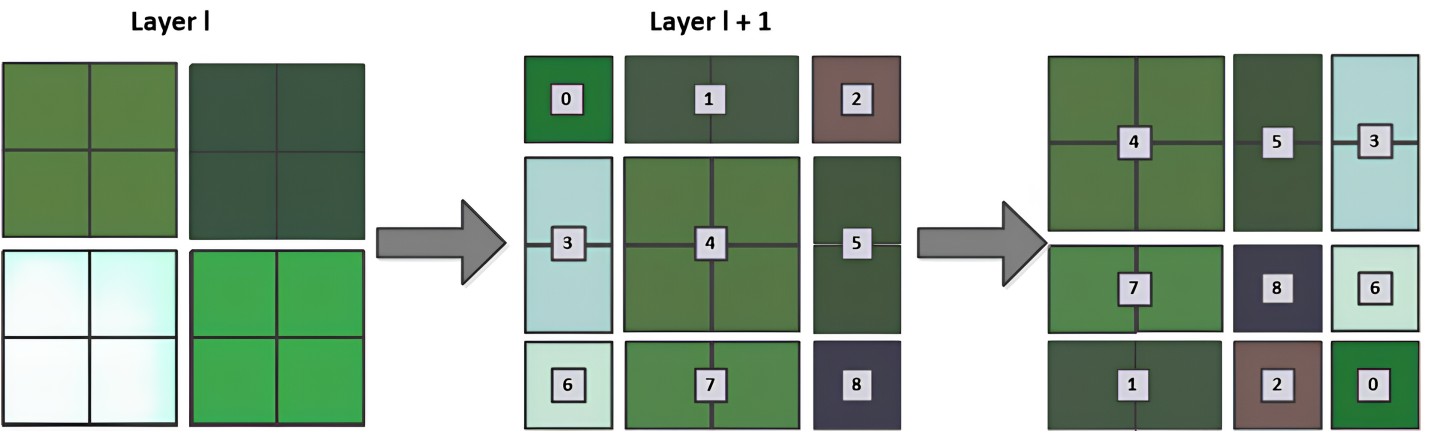

**Figure 5 SW-MSA calculation schematic.**

computational complexity. The W-MSA module conducts self-attention computations solely within its respective window, which may result in an inability to exchange information between windows. Therefore, the SW-MSA module is created. As shown in Fig. 5, the SW-MSA changes the relative positions of the image blocks, which is equivalent to re-segmenting the image blocks, and then calculates the self-attention between the windows. W-MSA and SW-MSA are generally used consecutively as a pair, if W-MSA is applied in the $l$-th layer, then SW-MSA is applied in the $(l + 1)$-th layer. Due to the window shift, in the $(l + 1)$-th layer, when determining self-attention for the windows, information exchange occurs with the windows from the $l$-th layer. This addresses the issue of inability to exchange information between different windows.

*Network construction process*
The Swin-T network is used to extract image tampering features, and its structure is illustrated in Fig. 6. Initially, in this article, the extracted features from spatial domain sensing module are input into the Swin-T network. These features are then divided into N non-overlapping patch tokens and concatenated along the channels to create a high-dimensional patch sequence. A Linear Embedding layer subsequently transforms the feature dimensions from 48 to a chosen dimension $C$. Subsequently, these processed features are fed into a self-attentive module (*i.e.*, Swin-T Block) with multiple sliding windows. After that, the height (H) and width (W) of each group with $2 \times 2$ adjacent patches are reduced by half, *i.e.*, transforming from H/4 × W/4 to H/8 × W/8. Meanwhile, the dimensionality of the patch tokens is doubled, *i.e.*, increasing from $2C$ to $4C$.

In each processing stage, the features pass through either a linear embedding layer or a patch merging layer and then through several pairs of attention modules with sliding windows, and thus constructing a hierarchical tampering feature representation. Such a design reduces the computational cost while reasonably utilizing self-attention to capture the correlation between different patches and better capture the tampering traces of the document images.

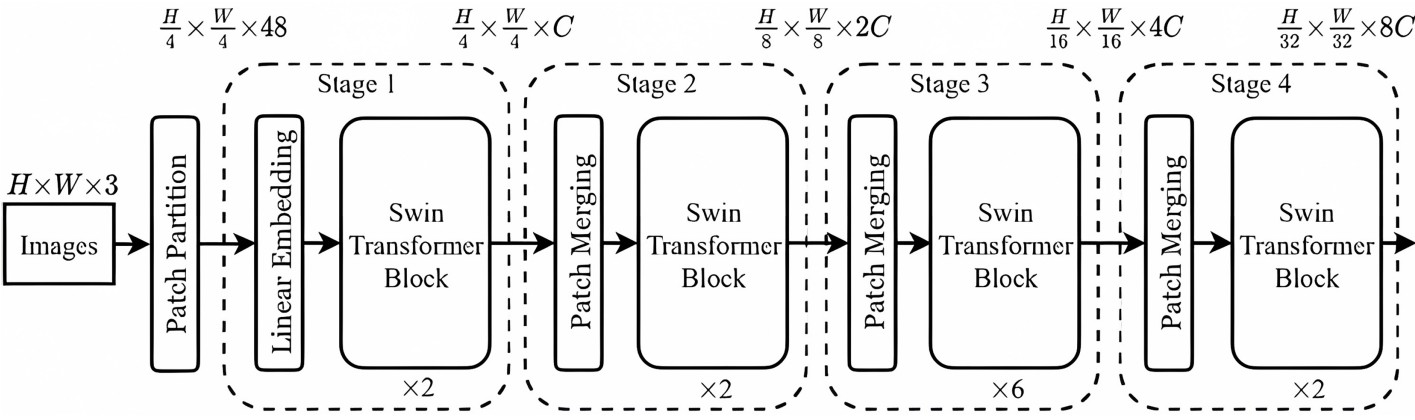

**Figure 6 Swin-T network structure.**

### Decoder network MRF-UperNet

As the backbone network deepens, the size of the image features diminishes, and the deep feature maps are rich in information but have low resolution, that make it more difficult to capture precise locations. In contrast, shallow feature maps contain clear positional information but have weak representational capacity, that make them unable to capture tampering information. To accurately detect the target location, fusing the semantic information from both deep and shallow layers is essential. Given that the proposed model adopts the structure of Swin-T, which possesses multilevel feature maps, this section introduces the pyramid structure of UPerNet (*Xiao et al., 2018*) as the architecture for the decoding network and incorporates the design of multiresolution fusion on the basis of this structure.

Essentially, multi-label classification for document image tamper detection is a semantic segmentation task, enabling the identification and localization of different tampered regions at the pixel level. In this article, the part of UPerNet network for region segmentation is selected and an improved UPerNet structure is designed. The primary structure of the original UPerNet is illustrated in Fig. 7A. UPerNet utilizes the standard Feature Pyramid Network (FPN) design, incorporating a top-down feature fusion pathway. FPN progressively passes and integrates high-level, semantically rich feature maps into lower-level, high-resolution feature maps, achieving multi-scale feature integration. It receives inputs from different scales extracted from the Swin-T backbone network and unifies the number of channels in the different layers by a $1 \times 1$ convolution. Beginning with the top layer of the backbone network (stage 4), the most profound features are extracted and processed through the Pyramid Pooling Module (PPM) to achieve multi-scale feature fusion $F_1$, and then upsampling it to the same size as that of stage 3 by $1 \times 1$ convolution, then perform the addition operation for feature fusion to obtain $F_2$. The similar process are performed on $F_2$ and $F_3$ to get the corresponding fused features. After that, the fused features $F_1$, $F_2$, $F_3$, and $F_4$ are combined using a $3 \times 3$ convolution and up-sampling with a uniform scale. The features are further fused to obtain information from

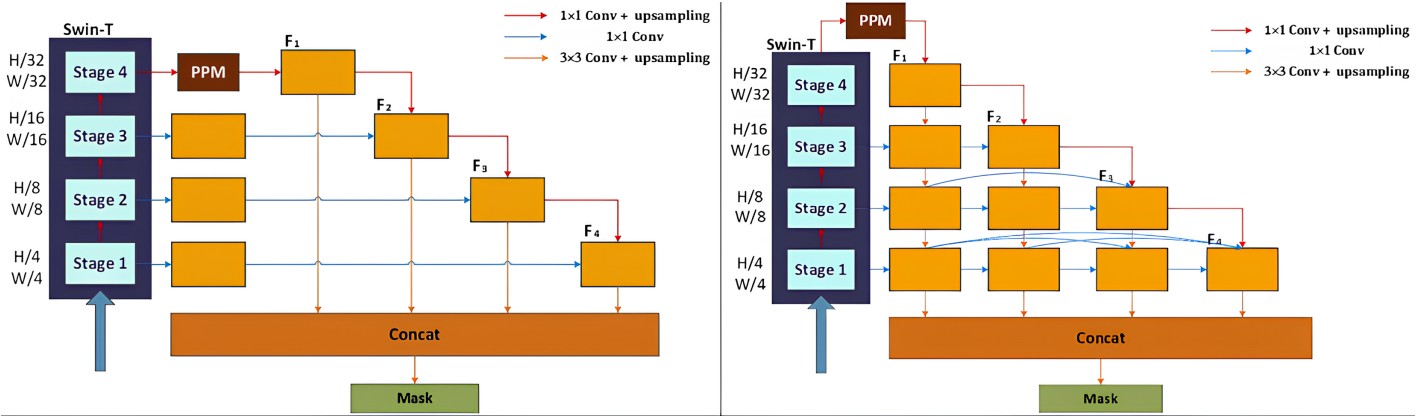

**Figure 7 Improved multi-resolution fusion network based on UperNet.**

different levels. PPM extracts global contextual information from feature maps through multi-scale pooling operations. This global information helps the network understand the overall structure of the scene, enhancing its perception of large-scale regions. PPM emphasizes global information, while FPN focuses on multi-scale, detailed, and local features. By concatenating the outputs of PPM and FPN and feeding them into the prediction module, the network can simultaneously utilize both global and local information, enhancing the accuracy and robustness of segmentation results. In this article, we enhance the primary structure of UPerNet by incorporating new convolution and fusion modules, as illustrated in Fig. 7B. $F_1$ is fused with the module extracted in stage 3 using $3 \times 3$ convolution and upsampling. This process is repeated on the subsequent modules to achieve continuous feature integration. Meanwhile, features of the same size in each stage are repeatedly utilized and continuously fused in each iteration through a $1 \times 1$ convolutional join. Therefore, we can obtain fully integrating information from multiple resolutions.

Additionally, we introduce a multi-resolution fusion network (MRF-UperNet) built upon the enhanced UPerNet, drawing inspiration from the human eye's repetitive observation process. Our goal is to detect traces of image tampering by continuously fusing high-resolution and low-resolution information. This design capitalizes on the richness of the network encoder's features at different resolutions, and also enhances the network's ability to perceive subtle tampering cues in images.

### Auxiliary detection head FCN module

During the backbone network's feature extraction process, the subsequent deep layers of the decoding network can lead to an increasing in the training difficulty and slow network convergence during joint training. Consequently, we propose a lightweight auxiliary detection head module, which adopts the basic fully convolutional network (FCN) (*Long, Shelhamer & Darrell, 2015*).

FCN is extensively utilized for semantic segmentation tasks. Generally, traditional convolutional networks add multiple fully connected layers following the last

convolutional layer to transform the convolutional feature maps into feature vectors for fixed-scale classification prediction, whereas FCN substitutes the fully connected layers with convolutional layers, allowing it to accept inputs of any size. The final feature map classification layer is upsampled using a $1 \times 1$ transposed convolutional layer, restoring it to the original image size to deliver pixel-level segmentation.

The auxiliary detection head module only works when the model is trained. During model validation and testing, only the base network combination of the backbone network and the encoder is used for the detection, and it isn't depend on the auxiliary detection head. Thus, only a few basic convolutional layers are employed to structure the output of the feature maps trained by the Swin-T backbone network. The primary details of the auxiliary detection head module are presented in Table 1, where Conv2d represents the convolutional layer, ConvSeg denotes the inverse convolutional layer, filters indicate the count of convolutions, *K* represents the count of kernels, *S* denotes the stride, *P* denotes the padding, *BN* denotes the batch normalization layer, and *ReLU* denotes the activation function. The input feature map for predictive classification has 512 channels. The input features are convolved once with a $3 \times 3$ kernel, then transforming them into values with 256 channels, and then an inverse convolutional layer is used for pixel-level classification.

The auxiliary detection header plays a significant role in the training process. It not only aids in the network's learning process but also enhances the model's sensitivity and accuracy in detecting tampering traces. Especially during the initial phase of network training, the auxiliary detection head module effectively mitigates the problem of vanishing or exploding gradients by providing additional gradient information. Therefore, it can accelerate the convergence of the network.

### Self-supervised augmentation and hard example mining strategies

#### Self-supervised augmentation

To address the challenge of limited mixed-type tampering samples in existing datasets, we developed a systematic self-supervised augmentation approach. Our method first applies conventional data augmentation techniques, including random vertical and horizontal flipping, random zooming, and image warping, to increase the basic dataset diversity.

For generating mixed-type tampering samples, we implement a sequential tampering process. We first preserve existing tampered regions in single-type tampered images, then systematically apply additional tampering operations. These operations include copy-move (duplicating content within specific regions), splicing (inserting external content), text tampering (generating synthetic text), erasing (removing content patches), and smearing (applying local blur effects).

To ensure the realism of generated samples, we apply several post-processing techniques including color adjustment to match surrounding regions, subtle noise addition, boundary smoothing, and contrast-brightness adjustment. Each augmented sample is accompanied by a detailed mask indicating the location and type of each tampering operation, including regions where different tampering types intersect.

**Table 1 Parameters of the auxiliary detection head module.**

| Layer name | Param |
| --- | --- |
| Conv2d | Filters: 256 K: $3 \times 3$, S: 1, P: 1 |
| BN | Channel 256 |
| ReLU | – |
| ConvSeg | Filters: 6 K: $1 \times 1$, S: 1, P: 1 |

This comprehensive augmentation process enables us to generate diverse mixed-type tampering scenarios while maintaining realistic visual appearance and providing accurate ground-truth annotations. Through this approach, the network can access more features of different tampering types during training, improving its ability to recognize and handle complex tampering situations.

*Hard example mining*
Image tampering detection algorithms typically suffer from imbalanced classes problem, where the real regions constitute a larger percentage from the entire image, while the tampered regions, especially the small forged regions that constitute a smaller percentage are difficult to be detect easily. However, these small tampered regions are often contain more important information. These areas are often the most difficult to accurately detect in a dataset. To address this challenge, we employ an online hard sample mining (OHEM) technique with a confidence threshold of 0.7 to identify and focus on challenging samples during training.

**Loss function**
The experiments in this article originally use a multiclassification cross-entropy loss function, which is crafted to ensure accurate prediction of both real and tampered regions. However, it does not fully consider the classification differences between different tampering types and the different effects when real and tampered regions are incorrectly predicted. During the training process, the gradient descent algorithm may fall into local optimal solutions, especially if the distribution of feature values is relatively sporadic, and the gradient of some features may disappear, thus significantly slowing down training speed.

For small object segmentation tasks, especially for images involving subtle tampering, the difference in distribution between different tampering types may be very small. This increases the risk of the model incorrectly predicting all regions as non-tampered regions, as it may be difficult for the model to accurately distinguish subtle tampering traces. To address this problem, we add Lovász-Softmax (*Berman & Blaschko, 2017*) loss to multiclassified cross-entropy, whose design concept is derived from a performance metric called the Jaccard index, also referred to as the IoU score. For the predicted value $M$ and true value $T$ of multiclass segmentation, the set of c-class wrongly predicted pixels is defined as $M_c$, which is shown in Eq. (2). The corresponding Jaccard loss $\Delta_{J_c}$ is shown in Eq. (3).

$$M_c(y^*, \tilde{y}) = \{T = c, M \neq c\} \cup \{T \neq c, M = c\}. \tag{2}$$

$$\Delta_{J_c} : M_c \in \{0,1\}^p \mapsto \frac{|M_c|}{|\{M = c\} \cup M_c|}. \tag{3}$$

The set of predicted pixels in the above equation is organized to obtain the expression of Lovász-Softmax for multi-class semantic segmentation as shown in Eq. (4), where $m(c)$ is the pixel error vector defined in Eq. (5), and $\phi_i(c)$ is denoted as the output score of the c-th class. The loss function obtained after Lovász-Extension smoothing is a continuous convex function, which is conducive to the optimization of the loss function.

$$L_{LS} = \overline{\Delta_{J_c}}(m(c)) \tag{4}$$

$$m(c) = \begin{cases} 1 - \phi_i(c) & \text{if } c = T \\ \phi_i(c) & \text{otherwise} \end{cases}. \tag{5}$$

The decoder network's loss function, $L_{decode}$, is determined as per Eq. (6), while the auxiliary detector head network's loss function, $L_{aux}$, is determined as per Eq. (7). Here, $L_{CE}$ stands for the cross-entropy loss, and $L_{LS}$ stands for the Lovász-Softmax loss.

$$L_{decode} = L_{CE_{decode}} + L_{LS_{decode}} \tag{6}$$

$$L_{aux} = L_{CE_{decode}} + L_{LS_{aux}}. \tag{7}$$

## RESULTS AND DISCUSSION

### Dataset construction and experimental setup

The scarcity of comprehensive document tampering datasets presents a significant challenge in current research. Most existing datasets are limited in both sample size and tampering diversity. For instance, the Tampered-SROIE dataset contains only 626 samples, which is insufficient for training deep learning models effectively. Moreover, existing datasets typically focus on single tampering methods without detailed type annotations, making multi-class tampering detection particularly challenging.

To address these limitations, we developed a new dataset by utilizing two source datasets: the StaVer Dataset and SCUT-EnsExam. The former comprises document images containing various elements such as colored logos and stamps, while the latter consists of handwritten exam sheets. We implemented a systematic processing pipeline to create our comprehensive tampering dataset:

First, we standardized all images to $512 \times 512$ pixels through random cropping. For the StaVer Dataset, we extracted three distinct regions from each image, while for SCUT-EnsExam, we selected ten regions per image. This initial processing resulted in a base collection of images suitable for tampering manipulation.

Our tampering generation process focused on five distinct manipulation types that commonly occur in real-world document forgeries:

The final dataset comprises over 30,000 images, with approximately 80% allocated for training and 20% for testing. Each tampering category is well-represented, with distribution as shown in Table 2. Figure 1 illustrates examples of each tampering type.

**Table 2 Statistics of the document tamper dataset (MixTamper).**

| Category | Types | Counts | Characteristics |
|---|---|---|---|
| Tampering types | Copy-move | 6,650 | Local duplication |
| | Splicing | 6,510 | External content insertion |
| | Text-generating | 6,678 | Artificial text addition |
| | Smearing | 6,300 | Local pixel distortion |
| | Erasing | 4,060 | Content removal |
| Dataset split | Training | 24,160 | 80% of total |
| | Testing | 6,038 | 20% of total |

Our dataset construction process followed systematic criteria to ensure diversity and minimize potential biases. We carefully selected source documents from both StaVer Dataset (containing colored logos and stamps) and SCUT-EnsExam (handwritten exam sheets), ensuring coverage of different document styles and content types. To maintain data quality and reduce potential biases, we implemented balanced sampling across tampering types (as shown in Table 2), with each type having approximately 6,000 samples except for erasing operations. The tampering operations were deliberately applied at varying locations and scales to ensure comprehensive coverage of different document regions and forgery patterns.

For comprehensive evaluation, we also incorporated the DocTamper dataset (Qu et al., 2023) as an external validation set. This dataset contains 170,000 document images in both Chinese and English, providing an independent benchmark for assessing our model's generalization capabilities. While DocTamper lacks detailed tampering type annotations, its diverse collection of document styles and languages makes it particularly valuable for validating our model's performance beyond the training distribution. The subsequent comparative experiments will be conducted on both our self-constructed dataset (hereafter referred to as MixTamper, as detailed in Table 3) and the publicly available DocTamper dataset to demonstrate the model's robust performance across different document scenarios.

Our dataset offers several key advantages: it provides a substantial number of samples, includes multiple tampering types with precise annotations, and represents both English and Chinese document scenarios. The tampering operations are designed to be subtle and localized, closely mimicking real-world forgery patterns. However, we acknowledge certain limitations: the document types are primarily confined to specific categories, and the fixed image dimensions may not fully represent all real-world scenarios.

For the experimental implementation, we employed the Adam optimizer with $\beta$ values of $(0.9, 0.999)$ and a weight decay rate of 0.01. The learning rate was managed using a poly strategy, initialized at 0.0001, with a linear warm-up period of 1,500 iterations. The training process incorporated both multicategorical cross-entropy and Lovász-Extension loss, weighted equally, over 300 epochs.

**Table 3 Ablation results for the backbone and decoder network structure on DocTamper dataset.**

| Network structure | P | R | F | IoU |
|---|---|---|---|---|
| U-Net | 86.44 | 75.11 | 80.38 | 67.2 |
| ResNet+UperNet | 86.36 | 68.24 | 76.24 | 61.6 |
| HRNet+UperNet | 89.66 | 68.00 | 74.72 | 66.29 |
| Swin-T+FPN | 88.63 | 71.76 | 87.30 | 77.39 |
| SwinT+UperNet | 90.88 | 89.66 | 90.27 | 82.26 |

## Visualization results

In order to show the prediction results more intuitively, tests are conducted on the MixTamper dataset and the corresponding prediction masks are generated. Some of the prediction results are selected as shown in Fig. 8, selected results are displayed from left to right: the tampered image (forgery), the ground-truth image (GT), and the prediction output of the proposed method (mask). In GT and mask, various colors are employed to distinguish between types of tampering: red for copy-move tampering (our work for copy-move forgery detection is only focus on the detection of target regions), green for splicing tampering, blue for text-generating tampering, yellow for smearing tampering, and purple for erasing tampering.

Figure 8 displays the experimental results. The first image contains a combination of five forgery types, which are challenging for existing state-of-the-art (SOTA) methods to detect. However, our proposed method can accurately identify and label each type, which demonstrates the model's excellent performance in dealing with multiple types of tampering, especially the erasure tampering, which is detected accurately by the algorithm even though the tampered area is very small. The second image contains two types of tampering: copy-move tampering and erasure tampering. The copy-move tampering moves the text below the basketball image to the right side of the number 2, which is relatively hidden, our algorithm is still able to accurately locate the tampering position. The third image contains two types of tampering: splicing tampering and text-generating tampering. Splicing tampering is more hidden, although its detection results are not clear at the tampering boundary, it is still able to accurately identify the general location, while text-generating tampering also achieves the desired effect. The fourth image involves copy-move, splicing and text generation tampering, all of which can be accurately detected. The fifth image incorporates four types of tampering, including copy-move, splicing, text generation and erasure tampering, in which the erasure tampering and text generation tampering means are intertwined, our method not only accurately recognizes each type of tampering, but also accurately locates the tampered area. Overall, our proposed model is capable of accurately identifying tampering types even in complex tampering scenarios.

## Ablation experiments and analysis

To evaluate the effectiveness of the proposed algorithm for detecting multiple types of forgery based on the Swin-T segmentation network in the spatial domain, an ablation

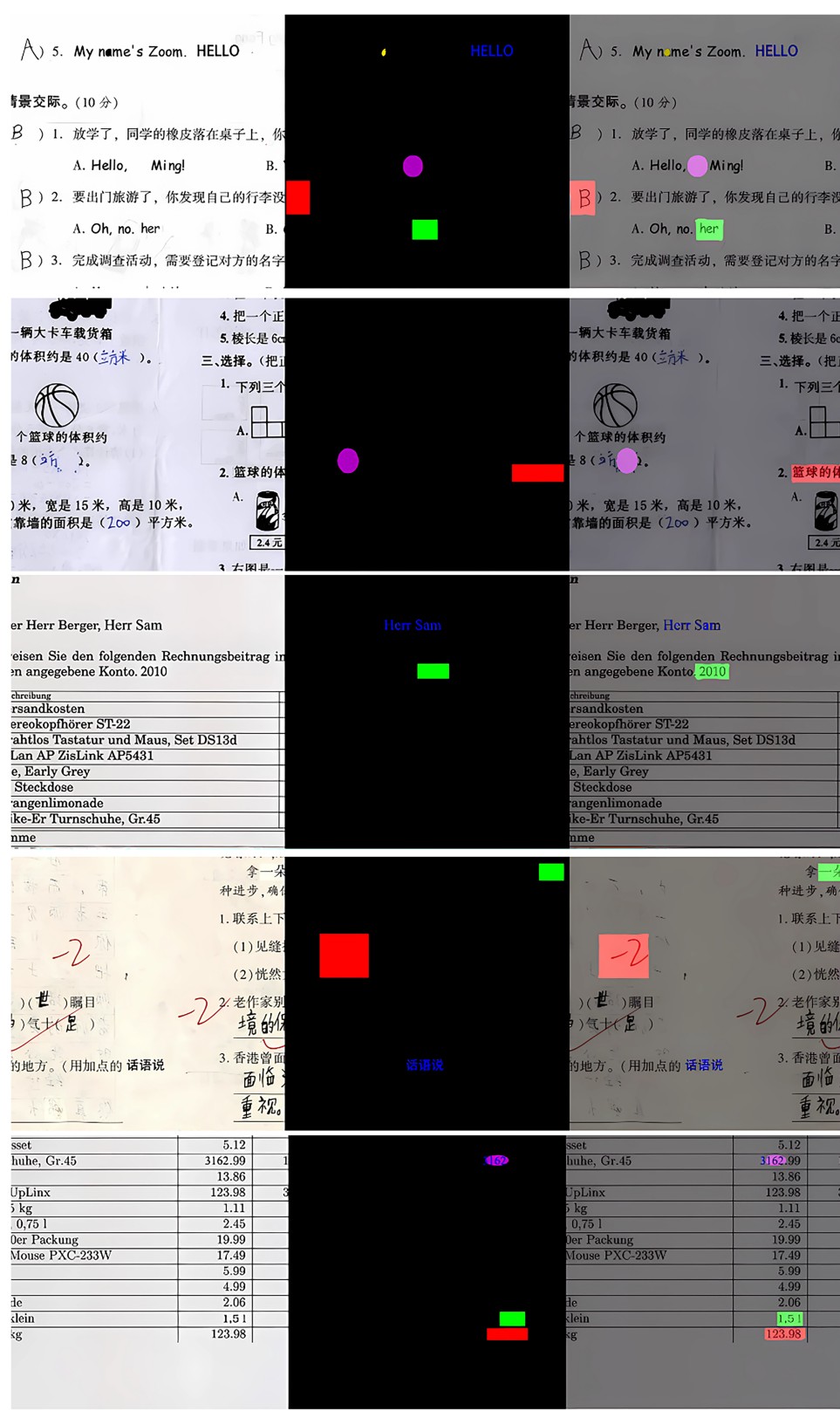

**Figure 8 Experimental visualization results for multiple forgery detection on MixTamper dataset.**

experiments are designed. On one hand, the performance of various backbone and decoder networks is compared, and detailed adjustments to the network structure are made. This allows for an analysis of how specific network structures impact the algorithm's performance. On the other hand, to verify the effectiveness of network optimization strategies, adjustments are made not only to the loss function but also by introducing hard sample mining and self-supervised augmentation methods. Corresponding ablation experiments are designed to support these analyses. Through these experiments, the positive effects of these optimization strategies in enhancing the performance of tamper detection algorithms are successfully verified.

### Ablation experiments based on network structures

Due to the data richness of DocTamper's dataset, it is more suitable for the experiments of the backbone network selection and it is also favorable for Transformer model training, which is utilized to perform ablation experiments on the network module. In this experiment, we employ backbone models such as U-Net, ResNet, and HRNet. Since U-Net inherently includes multi-scale information fusion, it does not require the addition of UPerNet. However, both ResNet and HRNet incorporate UPerNet as a decoder after feature extraction. Additionally, in the decoder module, we compare FPN network and UPerNet network through ablation studies. We thoroughly assess the network's performance using four metrics: precision (P), recall (R), F1-score (F), and intersection over union (IoU). The results of the ablation experiments are shown in Table 3.

From the comparison between the data in the first three rows and the fifth row, it can be seen that Swin-T's backbone network is improved in all four metrics compared with other backbone networks such as U-Net, ResNet, HRNet, *etc.* This could be attributed to Swin-T's ability to directly compute image correlations using the multi-head self-attention mechanism and acquire multi-scale features *via* the sliding window, enabling it to capture both local and global image features simultaneously. This enables better detection of tampering features. Meanwhile, from Table 3, the use of the UperNet network structure results in improvements across all four metrics compared to the FPN network structure. This is because UPerNet introduces richer contextual information and more effective feature fusion strategies, which enhance the network's adaptability to complex scenarios and various types of tampering.

In addition, based on Swin-T+UperNet structure, we further introduce the Spatial domain Perception Module (SPM) and the Multi-Resolution Fusion Module (MRF), and carry out ablation experiments on the self-built dataset MixTamper. These experiments extend the original dichotomous (0-1 segmentation) problem into a multichotomous (0-C segmentation) problem to cope with five different tampering types contained in the dataset: copy-paste tampering (copy-move), splicing tampering (splice), text-generation tampering (text), smearing (smearing) tampering, and erasure (erasure) tampering. The experiments are evaluated for these tampering types using IoU and F1 metrics, and the experimental findings are displayed in Table 4, from which it can be seen that the IoUs of copy-move and splice are improved by 3.77% and 3.18%, respectively, with the addition of the spatial domain perception module. This enhancement is due to the module's ability to

**Table 4 Ablation results for SPM and MRF module on MixTamper dataset.**

| Network structure | Copy-move | | Splice | | Text | | Smearing | | Erasure | |
|---|---|---|---|---|---|---|---|---|---|---|
| | IoU | F1 | IoU | F1 | IoU | F1 | IoU | F1 | IoU | F1 |
| w/o SPM | 69.15 | 81.86 | 75.91 | 86.36 | 77.79 | 83.27 | 88.75 | 94.03 | 93.17 | 96.47 |
| w/o MRF | 72.59 | 84.13 | 78.63 | 88.05 | 78.38 | 87.76 | 89.64 | 94.48 | 94.21 | 97.02 |
| Ours | 72.92 | 84.34 | 79.09 | 88.32 | 78.43 | 87.79 | 89.94 | 94.71 | 94.48 | 97.16 |

**Table 5 Ablation results for the network optimization module on MixTamper dataset.**

| Network structure | Copy-move | | Splice | | Text | | Smearing | | Erasure | |
|---|---|---|---|---|---|---|---|---|---|---|
| | IoU | F1 | IoU | F1 | IoU | F1 | IoU | F1 | IoU | F1 |
| w/o ADH | 67.76 | 80.91 | 74.16 | 85.23 | 77.56 | 87.25 | 88.37 | 93.34 | 92.54 | 96.27 |
| w/o HEM | 69.36 | 81.84 | 76.09 | 86.53 | 78.41 | 87.77 | 89.84 | 94.64 | 94.17 | 97.12 |
| w/o SSA | 71.44 | 83.38 | 77.53 | 87.37 | 78.20 | 87.65 | 89.39 | 94.25 | 93.92 | 97.11 |
| Ours | 72.92 | 84.34 | 79.09 | 88.32 | 78.43 | 87.79 | 89.94 | 94.71 | 94.48 | 97.16 |

extract richer information about the spatial domain, which often contains noise, edge details, and other potential tampering fingerprints. Meanwhile, the IoU for copy-move and splice also shows a slight improvement. This improvement is credited to the module for multi-resolution fusion's ability to effectively combine datas from both high and low resolutions, allowing the network to access richer information and make more precise tampering assessments.

### Ablation experiments based on network optimization strategies

We conducted comprehensive ablation experiments to evaluate three key components of our architecture: the auxiliary detection head (ADH), hard example mining (HEM) and self-supervised augmentation (SSA). Each component was evaluated for both performance metrics and training dynamics. Table 5 presents the ablation results.

The ablation results demonstrate the significant impact of each component:

The ADH improves both model performance and training efficiency. Without ADH, performance degrades across all tampering types (IoU drops of 5.16% for Copy-move, 4.93% for Splice). More importantly, ADH accelerates convergence—the model converges in 180 epochs with ADH compared to 245 epochs without it. This improvement stems from ADH's intermediate supervision signals, which provide better gradient flow during training.

The HEM significantly enhances the model's ability to detect challenging tampering cases. After introducing HEM, the IoU for complex tampering types (Copy-move and Splice) improved by 3.56% and 3% respectively. Moreover, HEM reduces training fluctuations and helps maintain stable convergence by focusing on informative samples.

The SSA provides consistent improvements across all tampering types (IoU increases of 0.23–1.48%). Beyond accuracy improvements, SSA enhances training stability and reduces

**Table 6  Analysis of component impact on model convergence.**

| Method | Mean IoU | Training epochs | Convergence loss | Training time |
|---|---|---|---|---|
| Baseline | 0.682 | 350 | 0.198 | 1.0× |
| With ADH | 0.715 | 300 | 0.185 | 1.2× |
| With HEM | 0.743 | 245 | 0.165 | 1.2× |
| With Both | 0.829 | 180 | 0.142 | 1.3× |

overfitting by introducing additional self-supervised tasks. The training loss converges more smoothly with SSA, indicating better feature learning.

To quantitatively evaluate the impact of the ADH and HEM on model convergence and training dynamics:

While Table 5 presents the detailed performance breakdown for each tampering type, Table 6 focuses on the convergence behavior of different model configurations. The results demonstrate the complementary effects of ADH and HEM on training dynamics, mean IoU is calculated as the weighted average across all tampering types. ADH improves convergence by providing intermediate supervision signals, reducing required epochs from 350 to 300 and improving mean IoU from 0.682 to 0.715. HEM further enhances optimization through focused learning on informative samples, achieving better convergence (loss of 0.165) with fewer epochs (245). When combined, these components show synergistic effects, achieving the best performance (mean IoU 0.829) with significantly faster convergence (180 epochs) despite a modest increase in per-epoch training time. This demonstrates that both components are essential for efficient model optimization and superior detection performance.

To determine the optimal configuration for our online hard example mining (OHEM) strategy, we conducted extensive parameter analysis experiments. We evaluated different confidence thresholds (0.5, 0.6, 0.7, 0.8, 0.9) and their impact on model performance:

As shown in Table 7, a threshold of 0.7 achieves the optimal balance between performance and training stability measured by loss variance in final 10 epochs. Lower thresholds (0.5–0.6) include too many easy samples, diluting the training focus, while higher thresholds (0.8–0.9) may overlook moderately difficult samples that are valuable for training. The selected threshold of 0.7 maintains approximately 32% of samples as hard examples, providing sufficient challenging cases while ensuring stable training.

## Analysis of combined loss function

To understand the effectiveness of our combined loss function, we conducted detailed experiments analyzing both individual components and their combination. The cross-entropy loss $L_{CE}$ focuses on pixel-wise classification accuracy, while the Lovász-Softmax loss $L_{LS}$ directly optimizes the IoU metric. We hypothesized that combining these losses would leverage their complementary strengths.

Our experiments reveal distinct characteristics and limitations of each loss function. As shown in Table 8, the cross-entropy loss ($L_{CE}$) shows rapid initial convergence but

**Table 7 Impact of OHEM confidence threshold on detection performance.**

| Confidence threshold | Mean IoU | Training stability | Hard sample ratio |
|---|---|---|---|
| 0.5 | 0.795 | 0.156 | 45% |
| 0.6 | 0.812 | 0.148 | 38% |
| 0.7 | 0.829 | 0.142 | 32% |
| 0.8 | 0.801 | 0.165 | 25% |
| 0.9 | 0.783 | 0.189 | 18% |

**Table 8 Training convergence analysis with different loss functions.**

| Epoch | $L_{CE}$ | $L_{LS}$ | $L_{CE} + L_{LS}$ |
|---|---|---|---|
| 20 | 0.685 | 0.592 | 0.701 |
| 40 | 0.723 | 0.634 | 0.758 |
| 60 | NaN | 0.651 | 0.792 |
| 80 | NaN | 0.663 | 0.815 |
| 100 | NaN | 0.670 | 0.831 |

**Table 9 Loss function ablation experiments.**

| Loss function | Copy-move | | Splice | | Text | | Smearing | | Erasure | |
|---|---|---|---|---|---|---|---|---|---|---|
| | IoU | F1 | IoU | F1 | IoU | F1 | IoU | F1 | IoU | F1 |
| $L_{CE}$ | 52.45 | 69.79 | 61.00 | 76.03 | 66.54 | 79.87 | 81.65 | 89.82 | 83.10 | 90.81 |
| $L_{LS}$ | 21.90 | 36.58 | 28.82 | 45.79 | 83.32 | 90.75 | 87.70 | 94.31 | 91.98 | 95.83 |
| $L_{CE} + L_{LS}$ | 72.92 | 84.34 | 79.09 | 88.32 | 78.43 | 87.79 | 89.94 | 94.71 | 94.48 | 97.16 |

becomes unstable after epoch 50, leading to NaN values. This instability arises because $L_{CE}$ struggles with the inherent class imbalance in tampering detection, where tampered regions often constitute a small percentage of the image.

The Lovász-Softmax loss ($L_{LS}$) shows slower but stable convergence throughout training. However, its final performance varies significantly across tampering types. While it excels at detecting tampering types with clear boundaries and uniform regions (Text: 83.32% IoU, Erasure: 91.98% IoU), it significantly underperforms on complex tampering types that require fine-grained feature discrimination (copy-move: 21.90% IoU, splice: 28.82% IoU), as displayed in Table 9.

Our combined loss ($L_{CE} + L_{LS}$ with 1:1 ratio) achieves both stable convergence and superior performance across all tampering types. As evidenced by Table 8, it maintains steady improvement throughout training while avoiding the instability of $L_{CE}$ alone. This improvement can be attributed to the complementary nature of the two losses: $L_{CE}$ provides strong pixel-level supervision that helps with fine detail preservation, while $L_{LS}$ optimizes the IoU metric directly, ensuring robust segmentation of

tampering regions regardless of their size. The combination particularly benefits complex tampering types, showing dramatic improvements in copy-move (72.92% *vs* 52.45% IoU) and splice (79.09% *vs* 61.00% IoU) detection compared to using either loss alone.

## Comparative experiments and analysis

To highlight the advantages of the proposed method, this section offers a detailed comparison of its performance with that of the existing RRU-Net (*Bi et al., 2019*), DenseFCN (*Zhuang et al., 2021*) and PSCC-Net (*Liu et al., 2021a*) for image tampering detection. To ensure comprehensive comparisons, we conducted comprehensive experiments on both our MixTamper dataset and the external DocTamper dataset.

For evaluation metrics, we use IoU and F1-score. For the DocTamper dataset, which only provides binary annotations, we compute these metrics directly on the binary prediction masks. For the MixTamper dataset with multi-class annotations, we calculate mean IoU (mIoU) and F1-score as follows:

The mIoU is computed by first calculating per-class IoU values and then averaging across all classes:

$$mIoU = \frac{1}{N} \sum_{i=1}^{N} \frac{TP_i}{TP_i + FP_i + FN_i} \tag{8}$$

where N is the number of classes ($N = 6$ in our case, including the background class), $TP_i$, $FP_i$, and $FN_i$ represent true positives, false positives, and false negatives for class i, respectively.

The F1-score is similarly computed per-class and then averaged:

$$F1 = \frac{1}{N} \sum_{i=1}^{N} \frac{2TP_i}{2TP_i + FP_i + FN_i}. \tag{9}$$

We use unweighted averaging across classes to ensure that each tampering type contributes equally to the final score, regardless of its frequency in the dataset.

As shown in Table 10, our method demonstrates consistent superior performance across both datasets. On the external DocTamper dataset, which contains 170,000 diverse document images in both Chinese and English, our method achieves an IoU of 84.16% and F1-score of 91.40%, significantly outperforming existing approaches. This strong performance on an independent external dataset confirms that our model's improvements are not overly specialized to our training set. Similarly, on the MixTamper dataset, we achieve an mIoU of 82.97% and F1-score of 90.46%, demonstrating excellent detection accuracy across different tampering types. The consistent performance between DocTamper and MixTamper (variance < 2% in both metrics) validates our model's robust generalization capability across different document styles and languages, while the significant margin over SOTA methods (improving IoU by >11% on both datasets) demonstrates the effectiveness of our approach.

**Table 10 Performance evaluation results on two different datasets.**

| Model | DocTamper | | MixTamper | |
|---|---|---|---|---|
| Indicator | IoU | F1 | mIoU | F1 |
| RRU-Net (*Bi et al., 2019*) | 37.77 | 50.74 | 36.38 | 50.08 |
| DenseFCN (*Zhuang et al., 2021*) | 40.34 | 51.25 | 39.66 | 50.43 |
| PSCC-Net (*Liu et al., 2021a*) | 72.26 | 77.08 | 69.39 | 80.34 |
| ours | 84.16 | 91.40 | 82.97 | 90.46 |

## Analysis of mixed-type tampering strategy

Most existing deep learning approaches for document tampering detection focus on binary localization without distinguishing between different tampering types. For instance, ObjectFormer, a state-of-the-art method, outputs a binary mask ($H \times W \times 1$) to indicate tampered regions and uses a classifier to determine authenticity. While effective for single-type tampering detection, this approach cannot effectively handle cases where multiple tampering types coexist. In contrast, our method treats tampering detection as a unified multi-class segmentation task, where each pixel is directly classified into one of six categories (original:0, copy-move:1, splicing:2, text:3, smearing:4, erasing:5).

To validate the effectiveness of our approach and justify the necessity of mixed-type tampering training, we conducted comprehensive experiments comparing ObjectFormer and our method under both single-type and multi-type scenarios. For fair comparison, we trained ObjectFormer on our dataset following its original configuration, while only considering localization accuracy (IoU and F1 score) in multi-type scenarios since ObjectFormer cannot predict specific tampering types. Table 11 presents the performance comparison:

The experimental results provide strong evidence supporting our mixed-type tampering strategy. In single-type scenarios (specifically copy-move tampering), ObjectFormer achieves slightly better performance, demonstrating its effectiveness in traditional tampering detection tasks. However, when handling multi-type tampering cases, ObjectFormer's performance drops significantly (IoU decreases by 15.8%), while our method maintains robust performance (only 1.4% decrease in IoU).

This performance gap can be attributed to two key factors: (1) Our unified multi-class segmentation approach preserves the spatial and semantic relationships between different tampering types (2) Training with mixed-type samples enables the model to learn complex feature interactions that commonly occur in real-world document forgeries.

These results clearly demonstrate that our mixed-type tampering strategy is not only theoretically sound but also practically beneficial. While existing methods like ObjectFormer excel in single-type scenarios, they struggle with real-world cases where multiple tampering types coexist. Our approach successfully addresses this limitation while maintaining comparable performance in traditional single-type detection tasks.

**Table 11  Performance comparison: ObjectFormer *vs.* our method.**

| Method | Scenario | IoU | F1 score |
|---|---|---|---|
| ObjectFormer | Single-type (Copy-move) | 0.843 | 0.915 |
| | Multi-type | 0.685 | 0.712 |
| Ours | Single-type (Copy-move) | 0.826 | 0.901 |
| | Multi-type | 0.812 | 0.885 |

**Table 12  Model performance under different image distortions.**

| Distortion type | Copy-move | Splice | Text | Smearing | Erasure |
|---|---|---|---|---|---|
| *JPEG compression (IoU %)* | | | | | |
| q = 90 | 61.65 | 70.98 | 77.94 | 69.85 | 73.37 |
| q = 80 | 54.35 | 59.34 | 77.48 | 60.33 | 56.50 |
| q = 75 | 51.79 | 55.72 | 72.10 | 53.17 | 51.10 |
| *Gaussian noise ($\sigma$)* | | | | | |
| $\sigma = 0.01$ | 65.82 | 72.45 | 75.89 | 71.23 | 75.41 |
| $\sigma = 0.02$ | 58.93 | 65.34 | 73.56 | 64.87 | 68.92 |
| $\sigma = 0.03$ | 52.76 | 58.91 | 70.23 | 57.45 | 61.34 |
| *Gaussian blur (kernel size)* | | | | | |
| 3 × 3 | 67.89 | 74.56 | 76.92 | 73.45 | 77.82 |
| 5 × 5 | 61.23 | 67.89 | 74.51 | 65.92 | 70.34 |
| 7 × 7 | 54.67 | 60.23 | 71.34 | 58.76 | 63.45 |

## Robustness analysis

To evaluate our model's robustness against real-world image distortions, we conducted comprehensive tests with various types of degradations:

Our analysis reveals that performance degradation varies significantly across different tampering types and distortions in Table 12. For JPEG compression, text tampering shows the strongest robustness, with only a 6.33% IoU decrease at q = 75, likely due to the high-contrast nature of text modifications. In contrast, Erasure tampering exhibits the most severe degradation (IoU drops by 43.38% at q = 75), as removed areas appear as uniform background regions that compression algorithms tend to oversimplify. Copy-move, splice, and smearing tampering show moderate but significant degradation (IoU drops of 21.13%, 23.37%, and 36.77%, respectively at q = 75), primarily due to compression artifacts masking fine-grained tampering traces.

Similar patterns emerge under Gaussian noise, where text tampering maintains relatively stable performance (7.54% IoU drop at $\sigma = 0.03$) due to its distinct contrast patterns. However, copy-move and splice detection suffer more severe degradation (20.16% and 19.18% IoU drops respectively at $\sigma = 0.03$) as noise obscures the subtle similarities between copied regions and spliced content. For Gaussian blur, the performance impact is generally less severe than compression or noise, but follows similar

patterns across tampering types. Text tampering remains the most robust (7.09% IoU drop at 7 × 7 kernel), while complex tampering types like copy-move and splice show greater sensitivity to the loss of high-frequency details (18.25% and 19.33% IoU drops respectively at 7 × 7 kernel).

Based on these findings, we propose several potential improvements for future research. These include incorporating distorted images during training to enhance model adaptation, developing more robust feature extraction mechanisms that preserve both local and global tampering evidence under distortions, and implementing adaptive detection thresholds based on estimated image quality. While implementing these improvements is beyond our current scope, they represent promising directions for enhancing the practical applicability of tampering detection systems.

## CONCLUSIONS

In this study, we present a multi-classification tamper detection algorithm utilizing Swin-T. This approach leverages the self-attention mechanism of Swin-T and the multi-resolution fusion decoder of the MRF-UperNet network for detecting multiple classes of image tampering. We propose multiple convolutional spatial domain perception module in the first layer of the network to enhance document tampering feature targeting. While in the network optimization, an auxiliary detection header is introduced to enhance the performance and an improved loss function is applied. Furthermore, a hard sample mining is implemented and a self-supervised augmentation is performed to expand the dataset. Experiments indicate that the proposed adjustments to the network structure and optimization measures significantly enhance the algorithm's convergence speed and detection accuracy. However, the proposed algorithm still suffers from some robustness problems, especially when facing image compression attack, the segmentation index decreases significantly, which limits the breadth and reliability of the tamper detection algorithm in the practical applications.

### Funding
This work was supported by National Natural Science Foundation of China (No. 62172132) and National Natural Science Foundation of China (No. 62471264). The funders had no role in study design, data collection and analysis, decision to publish, or preparation of the manuscript.

### Grant Disclosures
The following grant information was disclosed by the authors:
National Natural Science Foundation of China: 62172132, 62471264.

### Competing Interests
Ning Chu is employed by Zhejiang Shangfeng Special Blower Industrial Co.

## Author Contributions

- Li Li conceived and designed the experiments, performed the experiments, analyzed the data, prepared figures and/or tables, authored or reviewed drafts of the article, and approved the final draft.
- Kejia Zhang conceived and designed the experiments, performed the experiments, analyzed the data, performed the computation work, prepared figures and/or tables, authored or reviewed drafts of the article, and approved the final draft.
- Jianfeng Lu conceived and designed the experiments, analyzed the data, authored or reviewed drafts of the article, and approved the final draft.
- Shanqing Zhang conceived and designed the experiments, authored or reviewed drafts of the article, and approved the final draft.

## Data Availability

The MixTamper dataset is available at FigShare: zhang (2024). MixE.zip. figshare. Figure. https://doi.org/10.6084/m9.figshare.27633855.v2.

The DocTamper dataset is available at GitHub and Zenodo:

- https://github.com/qcf-568/DocTamper.

- dinmkel. (2025). DocTamper [Data set]. Zenodo. https://doi.org/10.5281/zenodo.14960484.

The StaVer Dataset dataset is available at Kaggle and Zenodo:

- https://www.kaggle.com/datasets/rtatman/stamp-verification-staver-dataset.

- Rachael Tatman. (2025). Stamp Verification (StaVer) Dataset [Data set]. Zenodo. https://doi.org/10.5281/zenodo.14957701.

The SCUT-EnsExam dataset is available at Github and Zenodo:

- https://github.com/SCUT-DLVCLab/SCUT-EnsExam.

- Huang, Liufeng and Chen, Bangdong and Liu, Chongyu and Peng, Dezhi and Zhou, Weiying and Wu, Yaqiang and Li, Hui and Ni, Hao and Jin, Lianwen. (2025). SCUT-EnsExam [Data set]. Zenodo. https://doi.org/10.5281/zenodo.14957677.

The code, including the configuration for building the entire network, the design of individual network structures (in mmseg), and the code used for training, is available in the Supplemental Files.

## Supplemental Information

Supplemental information for this article can be found online at http://dx.doi.org/10.7717/peerj-cs.2775#supplemental-information.

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
