# Peer review of "Multi-label classification for image tamper detection based on Swin-T segmentation network in the spatial domain"

_PeerJ Computer Science, doi:10.7717/peerj-cs.2775_

## Round 0.1 · original submission · Major Revisions

Dear authors,
You are advised to critically respond to all comments point by point when preparing an updated version of the manuscript and while preparing for the rebuttal letter. Please address all comments/suggestions provided by reviewers, considering that these should be added to the new version of the manuscript.

Kind regards,
PCoelho

Reviewer 1 ·

Basic reporting

The manuscript addresses multi-label classification of tampering types in document images using a Swin-T based segmentation network and several optimization strategies. While the idea is interesting, the paper contains several unclear points regarding the experimental design, the logic behind certain architectural choices, and the lack of certain analyses. Further clarity and additional experiments are needed to strengthen the methodology and support the conclusions.


1. The paper states that multiple tampering types can be accurately detected, but it is not clearly explained how the ground-truth masks were annotated for each forgery type or how the classification boundaries between these types were defined.
2. The description of the self-supervised augmentation process is insufficiently detailed, making it hard to understand the exact nature of newly generated mixed-type tampering samples and to confirm their realism.

Experimental design

3. The logic behind mixing multiple tampering operations in a single image is not well justified, and there is no experiment comparing performance on single-type vs. multi-type tampering scenarios to confirm that this mixing is beneficial.
4. The “hard example mining” step is introduced but not quantitatively analyzed; no experiments show how performance changes if hard example mining is omitted, making it unclear whether this component is truly necessary.
5. The selection criteria and potential biases for creating the MixTamper dataset remain unclear, and no external datasets are tested to confirm that the model’s improvements are not overly tailored to these particular training sets.

Validity of the findings

6. The paper reports strong results on two datasets, but the evaluation metrics for multi-class segmentation are not explained in detail, and it is not stated whether IoU and F1 are computed per-class and then averaged, or if some weighting strategy is applied.
7. The paper mentions that the model’s performance decreases significantly under compression attacks but does not provide any reasoning, ablation, or mitigation strategies, leaving the robustness claim partially unsupported.

Additional comments

8. The auxiliary detection head is claimed to improve convergence, but no ablation study is provided to confirm this claim, making the role of this component and its necessity in the final architecture unclear.

Reviewer 2 ·

Basic reporting

1. Highlight the unique aspects of your method in abstract.
2. Include a more detailed comparison with other Swin-T-based models to underscore the novelty (Introduction or related works).

Experimental design

3. Perform detailed analyses on varying parameter settings for hard sample mining and self-supervised augmentation techniques.
4. Extend robustness testing by simulating other real-world distortions, such as Gaussian noise, blurring, or varying resolutions.

Validity of the findings

5. Expand on the rationale behind using the Lovász-Softmax loss in conjunction with cross-entropy loss. Show how the combined loss impacts convergence and performance.
6. Ensure that the language is professional, clear, and concise throughout. Address any grammatical errors or typos.

Additional comments

7. Use consistent formatting for equations, figures, and tables for improved readability.

Please include all this improvement in your paper

---

## Round 0.2 · accepted · Accept

Dear authors, we are pleased to verify that you meet the reviewer's valuable feedback to improve your research.

Thank you for considering PeerJ Computer Science and submitting your work.

Kind regards
PCoelho

Reviewer 1 ·

Basic reporting

The manuscript addresses multi-label classification of tampering types in document images using a Swin-T based segmentation network and several optimization strategies.

Experimental design

The proposed model incorporating the Swin-T backbone with spatial domain sensing module shows thoughtful design choices targeted specifically at the challenges of tamper detection. The methods section provides sufficient technical detail and parameter specifications to enable replication.

Validity of the findings

The findings are well supported by comprehensive experimental evidence. The ablation studies systematically demonstrate the contribution of each proposed component, with clear quantitative improvements documented for the spatial perception module, multi-resolution fusion, auxiliary detection head, and optimization strategies.

Reviewer 2 ·

Basic reporting

Manuscript improved a lot

Experimental design

Manuscript improved a lot

Validity of the findings

Manuscript improved a lot

Additional comments

Manuscript improved a lot